# Mesoporous Nanoparticles for Diagnosis and Treatment of Liver Cancer in the Era of Precise Medicine

**DOI:** 10.3390/pharmaceutics14091760

**Published:** 2022-08-23

**Authors:** Han Wu, Ming-Da Wang, Jia-Qi Zhu, Zhen-Li Li, Wan-Yin Wang, Li-Hui Gu, Feng Shen, Tian Yang

**Affiliations:** 1Department of General Surgery, Cancer Center, Division of Hepatobiliary and Pancreatic Surgery, Zhejiang Provincial People’s Hospital, Affiliated People’s Hospital, Hangzhou Medical College, Hangzhou 310053, China; 2Department of Hepatobiliary Surgery, Eastern Hepatobiliary Surgery Hospital, Second Military Medical University (Naval Medical University), Shanghai 200438, China; 3Eastern Hepatobiliary Clinical Research Institute, Third Affiliated Hospital of Naval Medical University, Shanghai 200438, China

**Keywords:** mesoporous nanoparticles, liver cancer, precise medicine, tumor diagnosis, nanotechnology

## Abstract

Primary liver cancer is the seventh-most-common cancer worldwide and the fourth-leading cause of cancer mortality. In the current era of precision medicine, the diagnosis and management of liver cancer are full of challenges and prospects. Mesoporous nanoparticles are often designed as specific carriers of drugs and imaging agents because of their special morphology and physical and chemical properties. In recent years, the design of the elemental composition and morphology of mesoporous nanoparticles have greatly improved their drug-loading efficiency, biocompatibility and biodegradability. Especially in the field of primary liver cancer, mesoporous nanoparticles have been modified as highly tumor-specific imaging contrast agents and targeting therapeutic medicine. Various generations of complexes and structures have been determined for the complicated clinical management requirements. In this review, we summarize these advanced mesoporous designs in the different diagnostic and therapeutic fields of liver cancer and discuss the relevant advantages and disadvantages of transforming applications. By comparing the material properties, drug-delivery characteristics and application methods of different kinds of mesoporous materials in liver cancer, we try to help determine the most suitable drug carriers and information media for future clinical trials. We hope to improve the fabrication of biomedical mesoporous nanoparticles and provide direct evidence for specific cancer management.

## 1. Introduction

Liver cancer is the sixth-most-common primary cancer and the fourth-leading cause of cancer-related mortality worldwide [1,2]. Hepatocellular carcinoma (HCC) and cholangiocarcinoma (CCA) are the two most-common pathologic types of primary liver cancer, with HCC accounting for more than 80% of the total cases [3,4]. The most important causes of malignancy are liver fibrosis and inflammatory necrosis caused by chronic inflammation [5,6,7]. Although vaccination and antiviral therapy can effectively reduce the incidence of virus-associated liver cancer, the incidence of liver cancer due to chronic inflammation caused by metabolic liver disease and alcohol consumption is increasing year by year [8,9]. Due to the heterogeneity of the tumor gene, metabolism and immune microenvironment within liver cancer, the incidence of primary liver cancer remains high, while the diagnosis and treatment of HCC still face many challenges nowadays. Due to the insidious onset and rapid progress, most liver cancers are identified at the middle and late clinical stages [10,11,12]. Chemotherapy and targeting therapy can only slightly prolong the survival and improve the quality of life of patients, which are often unsatisfactory [13,14,15,16]. The limitations of diagnosis and treatment for liver cancer urgently need new integrated strategies to enrich and optimize existing management methods.

Mesoporous materials have been gradually developed as controlled drug-delivery carriers, since the discovery of their biological characteristics [17,18,19,20,21]. A variety of novel mesoporous nanoparticles have been prepared by controllable synthesis, with advantages such as larger surface area, higher pore volume, further modification of the surface and better biocompatibility [22,23,24]. With these unique drug-delivery properties, mesoporous nanoparticles play an important role in sustained drug release, targeting delivery for diseases in situ and maintenance of drug concentration in blood [25,26,27,28]. By designing the surface modifying protein and targeting peptide molecules with the help of fluorescent-dye guidance, mesoporous nanoparticles are revolutionizing the manner of personalized diagnosis and treatment of liver cancer [29,30,31]. Based on the design of most mesoporous nanoparticles, drugs or particles released locally at the liver cancer can accumulate in the tumor region and reach gradient concentrations. At the same time, with the help of advanced medical-imaging technologies, the aggregation of mesoporous nanoparticles can effectively inhibit liver cancer, while playing a therapeutic role, thus assisting in the rapid and accurate diagnosis of liver cancer [32,33].

In order to summarize the recent research on the delivery of drugs by mesoporous nanoparticles for liver cancer, we reviewed the literature on this topic and systematically summarized their points for further discussion. This systematic review mainly analyzes the properties of different mesoporous nanoparticles and the advantages and disadvantages of drug delivery from the perspective of different nanocomposites (Figure 1). Currently, the types of mesoporous materials mainly applied in the field of liver cancer include mesoporous silicon nanoparticles, ordered mesoporous carbon, mesoporous polydopamine, mesoporous bioactive glass, etc. In the diagnosis and treatment of liver cancer using mesoporous nanoparticles, the manuscript is mainly divided into four parts: accurate diagnosis, photothermal therapy, biological reactive oxygen therapy and liver fibrosis treatment. After summarizing the advantages and disadvantages of various kinds of mesoporous materials in the application of liver cancer, we hope to inspire researchers to explore more stable and effective ways to contribute to the diagnosis and treatment of cancer. It is believed that mesoporous nanoparticles can make new achievements in clinical translational research in the future, so as to benefit more tumor patients to a greater extent.

## 2. Mesoporous Nanocomposites

### 2.1. Silicon-Based Mesoporous Nanoparticles

Nanoparticles that are composed of silicon oxide or other silicon derivatives as mesoporous carriers have been the most widely used drug or molecular carriers in the field of nano-therapy for liver cancer [34,35,36,37,38]. Mesoporous nanoparticles, represented by mesoporous silica nanoparticles (MSNs), have a simple synthesis method and stable products with uniform particle size, which are applied as an excellent, available mesoporous biological medium [39] (Figure 2). However, common MSNs have poor biocompatibility, due to their tendency to coagulate or activate the complement system in the blood, even though they can protect the loaded drug to achieve a slow-release effect [40,41,42]. Then, hybrid mesoporous silicon nanoparticles have been developed, which make it easier for the body to degrade and remove mesoporous silicon after drug release, by hybridizing a variety of different atoms. In addition, in order to improve the loading efficiency of drugs, hollow mesoporous organosilicons (HMONs) have been designed and applied in the diagnosis and treatment of liver cancer. At the same time, because silicon cannot be directly used and metabolized by the human body, the design of mesoporous silicon should pay attention to the accumulation of elements in the liver and related systemic toxicity. Compared with inorganic silicon spheres, novel silicon nanoparticles effectively enhance the body’s metabolism of nanoparticles. The progress of the biological properties of such target products is constantly improving in adapting to the needs of the clinical translational application.

### 2.2. Ordered Mesoporous Carbon

Compared with other mesoporous particles, the pore size of ordered mesoporous carbon (OMC) is more uniform, and the distribution and arrangement of mesoporous carbon are more regular [43,44,45,46]. OMC is often used to design electrochemical signal amplifiers or immune signal sensors for biological detection, due to the electrical conductivity of regularly arranged carbon atoms [47,48]. In addition, because of the stability of carbon, OMC can be combined with other metal nanoparticles to detect the immunofluorescence signals of various proteins from blood samples. Nitrogen-doped mesoporous carbon has greater potential in the application of bioelectric signal amplification, which can contain more energy because of its excellent capacitance. In addition, OMC can be sequentially designed in different nano-stock forms for different biological detection purposes.

### 2.3. Mesoporous Bioactive Glass

Mesoporous bioactive glass is widely used in drug release in response to tumor microenvironment stimulation, due to its unique biodegradation properties [48,49,50,51]. Since bioactive glass can bind protons in the tumor microenvironment, it can quickly adjust the local tumor acidic microenvironment. In addition, bioactive glass has the function of an antibacterial effect and inducing angiogenesis, so it can be used for different biological application designs according to the properties of the different groups connected by silicon elements [52] (Figure 3). Similar to the properties of silicate nanoparticles, bioactive glass can synergistically play a variety of therapeutic effects on the basis of controlled drug release. Sol–gel-based strategies are one of the main methods to synthesize bioactive glass at room temperature. Metal–organic precursors and metal–salt precursors can be integrated into the glass network for therapeutic use in this way. In addition, bioactive glass nanoparticles can also be synthesized by the self-assembly of block copolymers. Doped with lanthanide ions, bioactive glass can be used for upconversion imaging for cancer diagnosis and drug monitoring.

### 2.4. Mesoporous Iron Oxide

Different from the elemental composition of the above-mentioned mesoporous materials, the iron elements of iron oxide enable it to have the potential of contrast agent imaging, except for drug loading [53,54,55]. Mesoporous iron oxide nanoparticles are widely used in the detection of serological substances due to their good biocompatibility and the ability to attach protein or peptide probes. At the same time, due to the magnetic properties of iron oxide, the intermediate pore iron oxide can be magnetically sorted in the production process [56,57]. In addition, in magnetic resonance imaging and magnetic thermotherapy, mesoporous ferric oxide has more advantages than other mesoporous carriers. In recent years, with the rise of chemodynamic therapy, the iron ion in mesoporous iron oxide, as a good catalyst of biological reactive oxygen, has been increasingly used in catalytic therapy [58,59]. The reaction mainly depends on the weak acidic microenvironment of the tumor and the Fenton-like catalytic reaction of the enzyme and iron oxide. Such a design for the tumor microenvironment can effectively avoid the toxic and side effects caused by the aggregation of mesoporous nanoparticles.

### 2.5. Mesoporous Polydopamine

Mesoporous polydopamine is often used as the surface coating of nanoparticles or self-assembled into separate nanoparticles for the delivery of tumor-inhibition drugs and the transmission of biological information [60,61,62]. Since dopamine itself is a common regulatory hormone in organisms, polydopamine has excellent histocompatibility, which other inorganic or organic mesoporous materials do not have. Due to this, mesoporous dopamine is designed for bionic surface modification and drug delivery, to take advantage of its resistance to easy immune recognition and elimination [63] (Figure 4). Its large surface area and masking effect allow nanoparticles such as iron oxide to be well-protected and to perform a magnetic resonance imaging function. In addition, the response of polydopamine to near-infrared light enables precise drug release and energy conversion. Such a combination of photo-response and photothermal therapy can be neatly integrated into the same nano-delivery system, which is a rare combination of material properties combining to create the ability to actively target tumor sites.

## 3. Diagnosis and Therapy of Mesoporous Nanoparticles for Liver Cancer

### 3.1. Accurate Diagnosis Induced by Mesoporous Nanoparticles

The traditional diagnosis of liver cancer relies on the detection of serum alpha-fetoprotein (AFP) and imaging methods such as ultrasound and magnetic resonance [64,65]. However, such medical tests are often hampered by limitations in serum sensitivity and by the degree of fibrosis in the liver itself [66,67]. In order to overcome the low sensitivity of serum detection, Shengzhong et al. used mesoporous carbon-bonded gold nanoparticles for AFP nanogram ultra-sensitive detection [68]. The immunological signal of AFP was transformed into the electrochemical signal by the sandwich structure of AuPt-MB and OMC@AuNPs. Due to the extensive and sensitive connection between AFP-Ab1 and Au-NPs, the electrochemical signal can be rapidly amplified. The sensitivity of AFP detection by this device is several times higher than that of conventional serum chemical methods.

In addition, the accurate diagnosis of liver cancer depends on the judgment of magnetic resonance imaging [69,70]. In order to increase the resolution of magnetic resonance imaging, mesoporous nanoparticles with different weighting modes have been designed and studied. Jiajia et al. used hollow mesoporous organosilica nanoparticles (HMONs) loaded with manganese atoms and FTCD gene plasmids to enhance the clarity of T1-weighted magnetic resonance imaging [71] (Figure 5). Compared with the gadolinium atomic imaging strategy, the biotoxicity of manganese-atom-doped HMONs was lower, and the T1-weighted imaging performance of MRI was not impaired. Meanwhile, synergistically loaded FTCD plasmids can effectively inhibit the activities of MMP and cytochrome C, by increasing the concentration of mitochondrial ROS, thus leading to the apoptosis of HCC cells. Kai et al. developed a novel strategy using mesoporous polydopamine loaded superparamagnetic iron oxide and mixed AFP with ferritin heavy chain (Fth) and sialic acid (SA)-PEI as active targeting tracers [72]. Since superparamagnetic iron oxide particles possess the ability to change the magnetic-resonance T2-weighted signal, the designed drug-delivery platform can be delivered to the liver cancer site with multiple targets. The T2-weighted imaging diagnosis performance of magnetic resonance can be effectively enhanced. Guang-Cong et al. loaded sorafenib onto bismuth-based nanomaterials and then coated the nanoparticles with polyethylene glycol and folic acid conjugations [73]. Due to the inclusion of bismuth, a good contrast agent for CT imaging, the nanosystem shows advantages over traditional contrast agents such as iohexol, not only in terms of biocompatibility.

In addition to the development of traditional imaging and serum diagnosis methods, mesoporous materials are more important in expanding new accurate diagnosis methods for liver cancer, in the context of the development of precision medicine [74]. Circulating tumor cells (CTCs) have become an important non-invasive method for the diagnosis of liver cancer in recent years. In addition to accurately determining the occurrence and type of liver cancer, CTCs help to understand the prognosis of liver cancer. Qihui et al. produced a CTCs capture system for HCC, consisting of coumarin-6, magnetic mesoporous silica nanoparticles and GPC-3 [30] (Figure 6). The GPC-3 molecule is used to enhance the collection capacity of magnetic mesoporous silica, while C6 converts the collected CTCs into the fluorescent signal. CTCs can be effectively distinguished from ordinary circulating cells by observing the fluorescence signal under a microscope and calculating its intensity. In addition, in order to break through the barrier of low resolution of traditional ultrasound imaging, Han et al. invented a new photoacoustic imaging agent. The photoacoustic imaging nano-contrast agent, which is based on mesoporous iron oxide and combined with the chemodynamic therapy of GOD, can increase the visualization rate of liver cancer to over 85%. More importantly, such a strategy can effectively remove the interference and influence of liver fibrosis background on imaging, while reducing the influence of operative factors.

### 3.2. Mesoporous Nanocomposites Mediated Local Therapy

The main methods for the local treatment of liver cancer using nanomaterials include local injection or paint of nanomaterials, photothermal ablation, photodynamic therapy and transhepatic arterial chemoembolization (TACE) [75,76,77]. Surgical resection can remove most tumor tissues and cells from the body, but the main problem is how to prevent the recurrence and metastasis of liver cancer after surgery. Bozhao et al. synthesized novel mesoporous silica nanoparticles, co-loaded with sorafenib and PD-L1 antibody [23]. Such nanoparticles are covered by platelet membranes, and PD-L1 antibody is attached on the surface of platelets. The use of this compound immune agent on the surgical wound surface of the mouse model of excised hepatocellular carcinoma can effectively prolong the survival period of the mice and prevent the recurrence of liver cancer. Yusheng et al. used pH-responsive hydrogel to bind mesoporous bioactive glass nanoparticles and DNase I, which was also used for wound resection after liver cancer surgery [52]. Inducing NK cell infiltration can effectively prevent the recurrence of wound liver cancer and has the advantage of low systemic toxicity.

Photothermal therapy is considered to be a very promising method for local tumor treatment because of its effective energy conversion and local temperature increasement [78,79,80,81]. For hepatocellular carcinoma that cannot be resected directly, photothermal therapy can become another model of local ablative therapy to supplement clinical application. Weijie et al. used mesoporous silicon to load IR780, a photothermal agent, coating nanoparticles with CAR T cell membranes [82]. Such a strategy is the first combination of CAR T immunotherapy with photothermal therapy for liver cancer. Zhenli et al. took the lead in mesoporous two-dimensional titanium carbide MXene surface, combining the photothermal performance of titanium carbide and mesoporous silicon loaded with chemotherapy drugs in a perfect way [83]. Jiayuan et al. utilized NIR-II light to modulate the release of DNase I and avoid liver metastasis by activating the immune system [63]. Jian et al. used gold nanoparticles as a medium for photothermal transformation to enhance the efficacy of HCC, in combination with the sustained release chemotherapy effects of mesoporous silica and sorafenib [84]. A similar strategy by Han et al. utilized ICG as a medium for photothermal conversion [85]. The common feature of the above methods is to skillfully combine energy conversion with synergistic drug release and systematic therapy to achieve the effect of amplification.

Compared with photothermal energy conversion, Shanyou et al. used TLS11a DNA aptamer as a photocatalyst to combine with BPQDs/Pt hybrid organosilicon for photodynamic therapy of liver cancer [86] (Figure 7). This strategy is a model for the local application of catalytic therapy for liver cancer. The nanoparticles of nucleus-targeting W18O49 were used by Da et al. to bind to mitochondrial-selective mesoporous silicon nanoparticles, simultaneously loaded with Ce6 as a photocatalyst [87]. This strategy effectively avoids heat damage to normal liver tissue and generates a large amount of singlet oxygen inside HCC cells, through highly selective nanoparticles. This effect can also induce the body to produce a long-term immune response, mainly including the up-regulation of immune factors and the increase in CD8+ T cells.

If the location of liver cancer is not suitable for local resection and ablation, TACE is an effective treatment to inhibit the growth of liver cancer. Li et al. used periodic mesoporous organosilicon and magnetic ferric oxide nanoparticles loaded with doxorubicin as arterial embolization agents [88] (Figure 8). Using Cy5.5 to trace the nanoparticles, it was found that HCC cells could effectively endocytosis the nanoparticles. At the same time, magnetic resonance could be used for real-time monitoring of hepatocellular carcinoma treatment. Such a strategy can serve as an effective complement to conventional chemoembolization when the traditional embolism methods fail, if the scenario and scope of their application are further optimized.

### 3.3. Mesoporous Chemical Systematic Therapy

In contrast to the local strategy, systemic therapy involves administering drugs intravenously or orally into the bloodstream [89,90,91]. Through the drug itself being aggregated in the liver cancer site or through the method of slow release to maintain the blood drug concentration, its purpose is to treat the liver tumor. These therapeutic ways include immunotherapy, targeted therapy and chemotherapy, and mesoporous materials serve as the carrier of these treatments, enabling drugs to be effectively protected and centrally delivered [92,93,94,95]. The first example we want to introduce is a novel gene-editing strategy in this area. Jing et al. used MSN as a carrier to co-package Cas 9 and RNA Ribonucleoprotein complex (RNP) into liposomes to edit genes such as PCSK9, APOC3 and ANGPTL3 [96] (Figure 9). Such techniques achieve gene-editing efficiencies of more than 50% in living livers. More importantly, such a technique could be extended to other mutated diseases, in addition to inactivating some mutated fat-metabolism genes. Moreover, Yinxing et al. effectively used MSN loaded with shRNA to regulate the expression of GPC3 [97]. Tanshinone IIA was added as a therapeutic adjuvant and co-released in the HCC region. ShRNA, a common tool for regulating cellular gene expression, can maintain the long-term down-regulation of specific gene expression and affect its function. In addition, the aforementioned imaging strategies also include the regulation of gene expression through FTCD plasmid. Therefore, epigenetic means of gene expression regulation can be effectively applied in the treatment of liver cancer.

Compared with the improvement of gene-editing efficiency, the delivery and sustained release of chemotherapy drugs is another important development direction of mesoporous materials in the treatment of liver cancer. Xiaoqin used large-pore mesoporous silicon to achieve targeted delivery of arsenide Trioxide (ATO), which is a recognized solid tumor-treatment drug [98]. This strategy combines prodrug delivery and targeting ligand folic acid to enhance the local efficacy of the drug, while reducing adverse effects on normal tissues. Sorafenib, as a first-line systematic treatment for liver cancer, has also been widely used in the strategy of mesoporous drug delivery [35,99,100,101]. The strategy of Han et al. may enable these kinase inhibitors to be more effective in the localized HCC [85]. Doxorubicin (DOX) is another typical application of systemic therapy. Due to its broad-spectrum anticancer effect, there is also evidence that doxorubicin can mediate immune-related apoptosis. Zheng et al. applied CT as a navigation and monitoring tool to effectively combine gold nanoparticles and DOX to form an effective tumor DNA-breakage strategy [74]. This strategy organically combines radiotherapy and chemotherapy and forms a promising comprehensive platform. Nan et al. combined DOX with high-intensity focused ultrasound, and hollow mesoporous Prussian blue nanoparticles became the perfect carrier of such a combined strategy [102]. Such designs tend to have more diverse therapeutic targets than gene editing and, thus, have non-specific anticancer effects.

Immunotherapy through systemic circulatory system has not been effectively developed in the design of mesoporous materials for liver cancer [103,104,105]. However, the neoantigen design mentioned earlier converts the cold tumor of liver cancer into a hot tumor, meaning that the immune cells in the tumor are infiltrated and active. Activation of such systemic immune effects is achieved by photodynamic therapy against HCC. Similarly, the combination therapy design of Han et al. mentioned earlier incorporates photothermal therapy as a priming treatment for immune antigen release, to enhance the immune response. We attempted to summarize the above studies on the use of mesoporous nanoparticles for in vivo therapy and analyzed the similarities and differences (Table 1). In fact, these strategies for ablating tumors in situ, killing HCC cells and inducing antigen release may have similar immune effects. In addition, in the design of immune-system therapy in the future, researchers could try to activate systemic anti-tumor immunity by directly loading immune agents with mesoporous materials.

### 3.4. Prevention of Liver Cancer by Inhibiting Liver Fibrosis

In recent years, with the deep understanding of the microenvironment of chronic liver inflammation, chronic liver fibrosis with multiple cell participation, inflammatory factor mediated and liver stromal cell gene mutation is one of the main etiological causes of primary liver cancer [106,107,108]. The way to effectively inhibit the infiltration and proliferation of inflammatory cells improves the fibrosis degeneration of hepatic stellate cells and regulates the relevant inflammatory pathways has become another important measure to prevent liver cancer [109,110]. After the in-depth understanding of the related mechanisms of cirrhosis, researchers selected appropriate mesoporous materials to be applied in the process of anti-fibrosis. Qianjun et al. designed SBA-15 and rhodamine B to shape the surface area and pore size of MSNs for loading the negatively charged drug salvianolic acid B [111]. With the release of the loading drugs, intracellular ROS levels are effectively inhibited, so this strategy can effectively inhibit intracellular lipid peroxidation and associated inflammatory responses, thereby reversing hepatocyte fibrosis. Juan L et al.’s strategy was to target the tenascin-C (TnC) protein secreted by activated hepatic stellate cells, which silenced the TnC protein by siRNA, to reduce the level of extracellular inflammatory factors and liver cell migration [112] (Figure 10). Regulating the physiological behavior of liver cells by influencing the microenvironment of liver cells is an important aspect of preventing the occurrence of liver cancer.

In addition to the above biomedical approaches, the herbal extract costunolide (COS) has also been found to be effective in inhibiting liver fibrosis. Xia et al. used pH-responsive MSNs to attach to COS, the main design concept of which was the unstable release of methacrylic acid copolymer in acidic solution [113]. This strategy effectively overcame the dilemma that COS would be degraded by gastric acid by direct oral administration and effectively enhanced the efficacy of COS. In another study to improve the bioavailability of oral drugs, the authors synthesized N-(3,4,5-Trichlorophenyl)-2(3-nitrobenzenesulfonamide) benzamide as the loading material of MSNs [114]. As a result, the levels of α-SMA, TGF-β1 and MMP-2 were down-regulated for the further treatment of liver fibrosis.

Ayman et al. found that MSNs have significant hepatorenal toxicity, which may itself cause liver function and histological changes [115]. The levels of ROS, intracellular lipid peroxidation, nitric oxide, antioxidant inhibition and Nrf2/HO-1 signal transduction may be increased by the integrated MSNs’ application. This is a warning for researchers who simply use bare MSNs, and it suggests that researchers should pay great attention to the biological toxicity of inorganic materials themselves and try to reduce the effects of materials on liver function. Major strategies include modifying the physical and chemical properties of materials such as MSNs by surface modification or hybridization, loading mild drugs to neutralize the toxicity of inorganic materials or using anti-inflammatory substances to eliminate the body’s stress response to substances such as MSNs. In conclusion, inhibiting liver fibrosis should not be ignored as an important means of preventing liver cancer in the development of mesoporous biomaterials, but the pro-fibrosis effect of some materials should be alleviated before application.

## 4. Discussion and Outlook

As a kind of superior biomaterials, mesoporous nanoparticles have been widely used in the diagnosis and treatment of liver cancer in recent years [52,116,117,118]. Due to their physical and chemical properties such as large surface area, controllable pore size and controllable degradation, they are a good carrier for drug therapy [119,120]. In order to enhance the fit between mesoporous materials and the diagnosis and treatment of liver cancer, previous studies have taken the initiative to integrate nano-contrast agents, immune microenvironment regulation and gene editing into the design of mesoporous materials combination therapy. Similar to hollow materials, membrane-wrapped materials and polymer materials, the design philosophy of such studies is often guided by the principles of targeted delivery and the slow release of drugs [121,122,123]. It is because of so many different attempts that mesoporous materials constantly overcome the shortcomings of low biocompatibility and a high miss rate, so gradually they can be used in the systemic and local treatment of organisms.

Although the development of mesoporous nanoparticles is very rapid, there are still some limitations and deficiencies in biological applications. The disadvantages are mainly reflected as follows: (1) Some monomer mesoporous nanoparticles such as MSNs have liver and kidney toxicity, so they cannot be directly applied to the treatment of liver cancer and other tumors. Methods must be used to reduce or neutralize the toxicity of the material. In addition, because the source and mechanism of toxicity of materials are not clear, researchers need to further think about and discover the causes of organ dysfunction. In addition, as a large number of nanoparticles accumulate in the liver, the primary site of HCC, in order for them to play a role, it is necessary to pay attention to the elemental toxicity of these mesoporous nanoparticles and the systemic toxicity caused by their aggregation. (2) Many researchers have ignored the comprehensive characterization of the properties of mesoporous materials in order to pursue comprehensive strategies of drug delivery and combination therapy. This is mainly reflected in the insufficiency of electron microscope observation, the lack of solution stability and agglomeration experiment, the lack of property verification after drug release and the limitation of in vitro verification of biological reactive oxygen species. (3) Due to the particularity of the tumor microenvironment of liver cancer, its blood supply characteristics are inconsistent with those of other malignant tumors, so the immune cells and inflammation are also different. Therefore, it is necessary for more researchers to conduct experiments on tumor models in situ of the liver, in order to draw meaningful conclusions. In fact, due to the strong nanoparticle adsorption and drug removal effect of the liver itself, many nanocarriers are often unable to accurately reach the tumor site delivery. This is also one of the main reasons for the low efficacy and drug tolerance of some new drugs in clinical development. (4) Last but not least, a large number of studies on mesoporous materials only focus on the loading and release of drugs and do not pay attention to the different physical and chemical properties and biological properties of different mesoporous materials when they are used as carriers to target tumors. This needs to be further improved in future studies.

In the era of precision medicine guiding scientific research and clinical practice, the research and development of mesoporous nanoparticles is becoming more and more targeted and individually adaptable. As the context of systematic treatment tends to be microscopic and diversified, it is believed that mesoporous materials can contribute more strategies to the diagnosis and treatment of liver cancer in tissue engineering, tumor precision therapy, immunology and other fields. At the same time, in the field of real-time tumor monitoring and early screening of diseases, mesoporous nanomaterials also shoulder a great mission, so the advantages of these tasks are due to mesoporous nanomaterials’ strong ability of multi-molecule integration. In the future, the main development directions of mesoporous nanomaterials in the field of liver cancer and even cancer diagnosis and treatment may be as follows: (1) more biomimetic and biocompatible mesoporous particles as nano-drug carriers; (2) more efficient energy and image transformation to facilitate the identification of tumor regions and metastases; and (3) more precise and targeted development of new antigens and related chemical groups. In the future comprehensive diagnosis and treatment of liver cancer, it is believed that more and more mesoporous nanomaterials will be applied to clinical trials and precision medicine.

## Figures and Tables

**Figure 1 pharmaceutics-14-01760-f001:**
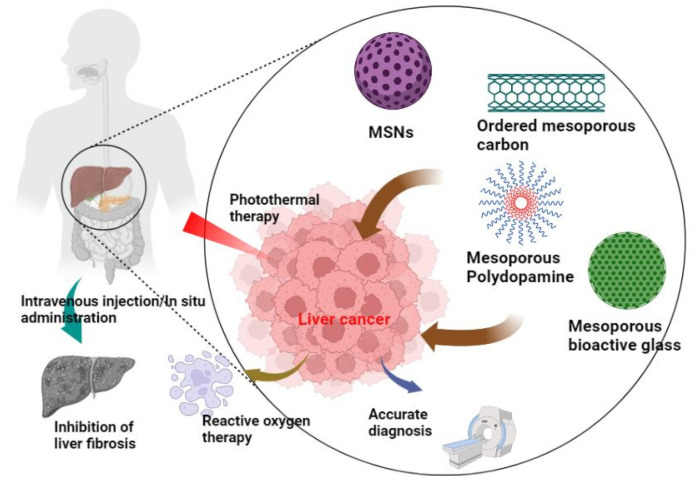
Summary scheme of mesoporous materials for the diagnosis and treatment of liver cancer. Currently, the types of mesoporous materials applied in the field of liver cancer mainly include mesoporous silicon nanoparticles, ordered mesoporous carbon, mesoporous polydopamine, mesoporous bioactive glass, etc. The biological applications of mesoporous materials include accurate diagnosis of liver cancer, systematic and local treatment and the treatment of cirrhosis.

**Figure 2 pharmaceutics-14-01760-f002:**
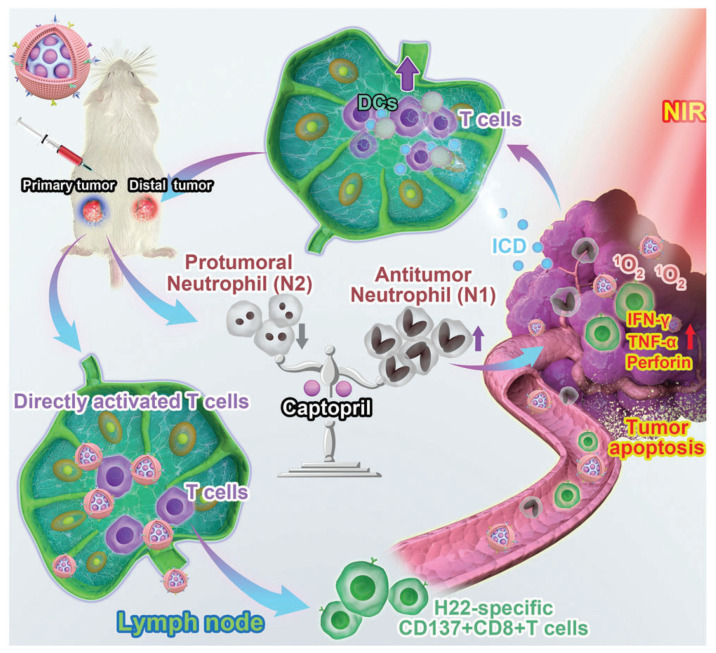
Illustration of an example using well-designed mesoporous silicon as an immunotherapy vector for liver cancer. Captopril is the cargo of MSNs for targeted delivery to lymph nodes, while DCs membranes are highly selective envelopes. Through the protection of MSNs and the targeting of DCs membranes, captopril release induces neutrophil differentiation and infiltration, thus achieving the regulation of tumor local immune microenvironment. Reprinted with permission from Ref. [39]. Copyright 2022 John Wiley and Sons.

**Figure 3 pharmaceutics-14-01760-f003:**
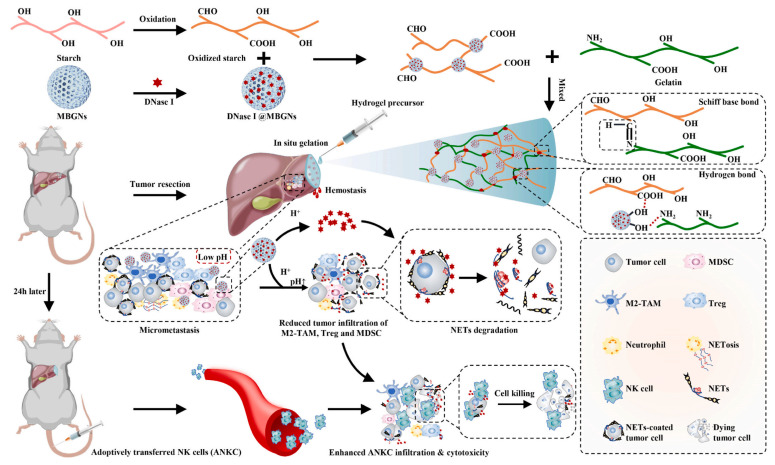
Schematic diagram of mesoporous plexiglass loaded with hydrogel as the drug carrier and the polymer binder. Such a scheme subtly covalently binds MBGNs and polymers by carboxyl groups of gelatin and starch with hydroxyl groups on mesoporous surfaces. This strategy can be applied to the wound of hepatocellular carcinoma surgery in situ, and NK cells are the main target cells and activation sites. Reprinted with permission from Ref. [52]. Copyright 2022 Elsevier Ltd.

**Figure 4 pharmaceutics-14-01760-f004:**
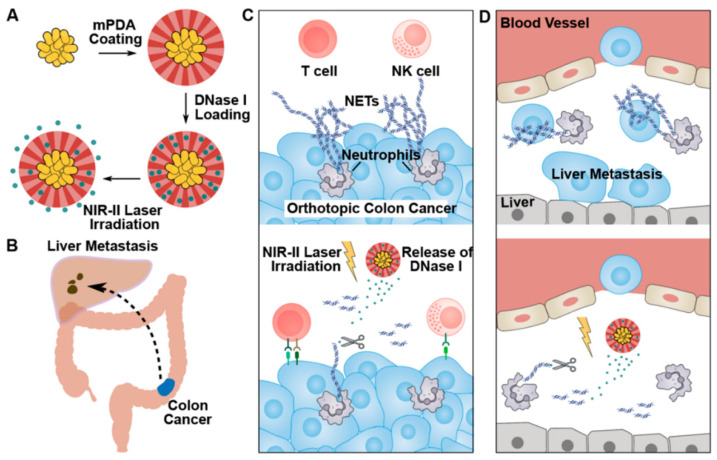
Synthesis and schematic diagram of dopaminergic and gold nanoparticles loaded with DNase for cancer therapy. (**A**,**B**) The process of DNase I delivery and liver matastasis of colon cancer. (**C**,**D**) Biodegradation and anticancer effect of synthesized system. The strategy reveals the movement process and delivery mode of nanoparticles and illustrates the activation of immune cells. Reprinted with permission from Ref. [63]. Copyright 2022 ACS publications.

**Figure 5 pharmaceutics-14-01760-f005:**
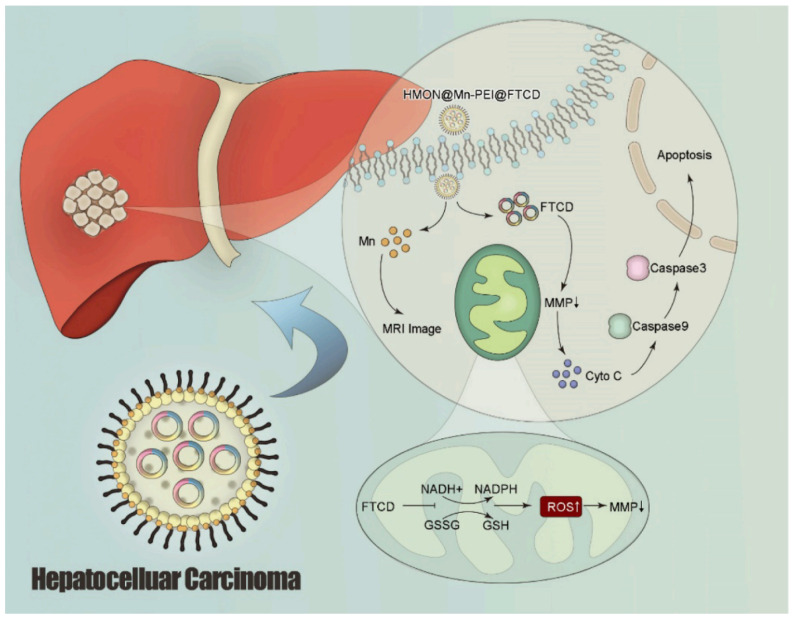
Schematic diagram of the mechanism of HMON-loaded FTCD nanoparticles in hepatocellular carcinoma, by activating mitochondrial apoptosis signaling pathway. Reprinted with permission from Ref. [71]. Copyright 2022 Elsevier Ltd.

**Figure 6 pharmaceutics-14-01760-f006:**
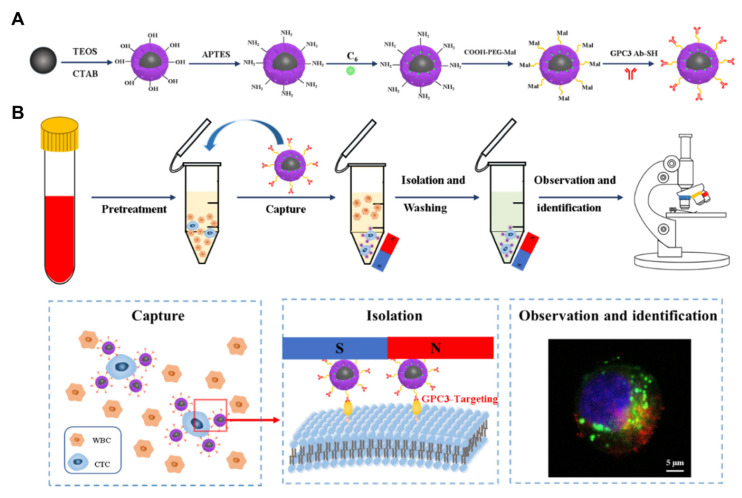
A method for highly specific capture, isolation and observation of hepatocellular carcinoma CTCs by immunomagnetic fluorescence nanodevices targeting GPC3. (**A**) The synthesis of immunomagnetic fluorescence nanodevices. (**B**) The capture principle of CTCs with immunomagnetic fluorescence nanodevices. The figure shows the synthesis process of nanoparticles and the capture and recognition process of CTCs. Reprinted with permission from Ref. [30]. Copyright 2021 Dovepress.

**Figure 7 pharmaceutics-14-01760-f007:**
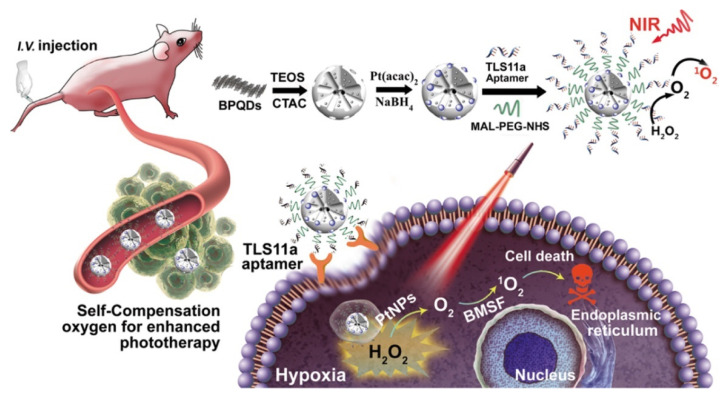
Case diagram of a mesoporous silicon metal framework using black phosphorus quantum dots for photodynamic therapy. Reprinted with permission from Ref. [86]. Copyright 2019 ACS publications.

**Figure 8 pharmaceutics-14-01760-f008:**
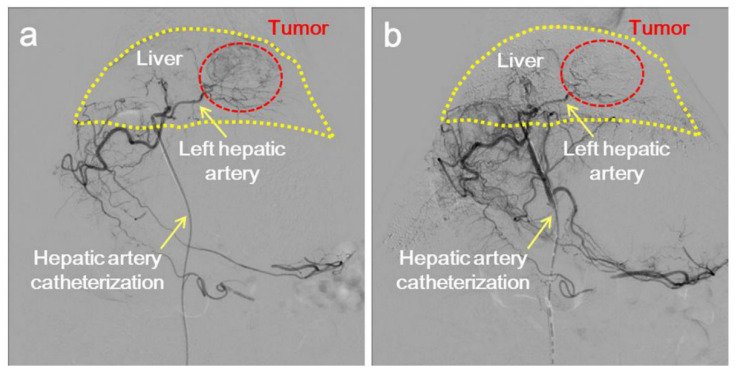
Integrated periodic mesoporous organosilicon loaded ferric oxide particles and doxorubicin for TACE treatment of liver cancer. (**a**,**b**) The comparison of particle induced TACE before and after treatment. The schematic diagram reveals the hepatic artery angiography with DSA guidance. Reprinted with permission from Ref. [88]. Copyright 2021 Elsevier Ltd.

**Figure 9 pharmaceutics-14-01760-f009:**
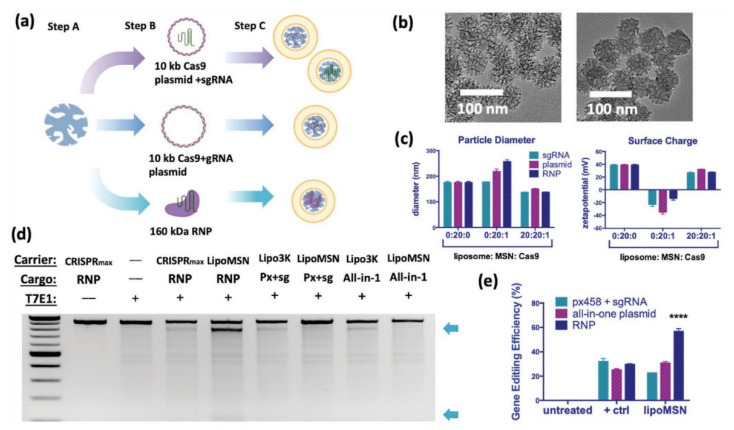
In vitro design and optimization diagram of CRISPR delivery and gene editing in liver, using MSN coated with liposome as a carrier. (**a**) The liposome loading capabilities of gene editing substance. (**b**–**e**) The characteristics of CRISPR delivery system **** *p* < 0.0001. Reprinted with permission from Ref. [96]. Copyright 2020 John Wiley and Sons.

**Figure 10 pharmaceutics-14-01760-f010:**
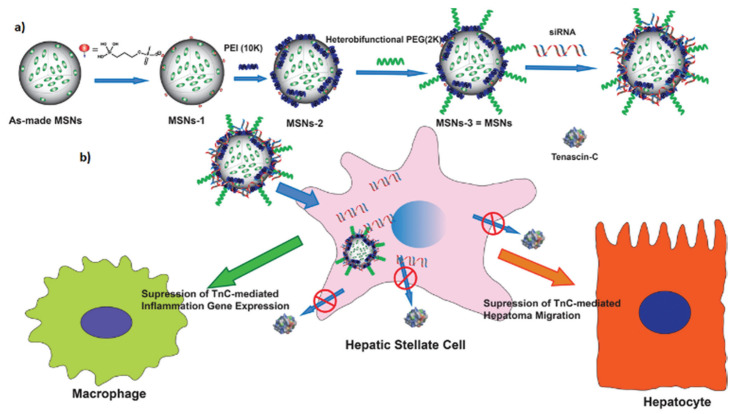
Synthesis of PEG/PEI-coated monodisperse silica microspheres and schematic diagram of siRNA-MSNs carrying, transporting and delivering siTnC and down-regulating TnC protein in hematopoietic stem cells. (**a**) The PEI coated MSNs synthesis. (**b**) The regulation of TnC expression by nano-delivery. Reprinted with permission from Ref. [112]. Copyright 2019 The Royal Society of Chemistry.

**Table 1 pharmaceutics-14-01760-t001:** Summary of mesoporous nanoparticles application in vivo for HCC.

*Category of Mesoporous Nanoparticles*	*Loading Cargos*	*Animal Models*	*Biological Anti-Tumor Effects*	*References*
Hollow mesoporous organosilica	Mn ions, FTCD plasmids	Nude mice	Activating the mitochondria-mediated apoptosis signalling	[71]
Mesoporous polydopamine	SPIO, AFP-Fth	BALB/c mice	Targeting, MRI T2-weighted imaging	[72]
Hollow mesoporous silica	Anti-PD-L1 antibody	C57BL/6J mice	Increasing CD4+ and CD8+ T cell populations	[76]
Mesoporous silica coated with CAR-T cell membranes	IR780	BALB/c-nu mice	Photothermal antitumor with enhanced targeting abilities	[82]
Co-loaded mesoporous silica nanosystem	Indocyanine green (ICG), sorafenib	C57BL/6J mice	Photothermal tumor killing effect, immune-enhancement capability	[85]
Periodic mesoporous organosilica	Magnetite Fe_3_O_4_ nanoparticles, Cy5.5	ICR mice	Transarterial embolization	[88]
Liposome-coated mesoporous silica	Cas9 plasmid, Cas9 protein/guide RNA ribonucleoprotein complex	C57BL/6J mice	Efficient, combinatorial gene-editing therapeutics	[96]
Mesoporous polydopamine	Sorafenib, ultrasmall SPIO nanoparticles	Nude mice	PTT boosting the ferroptosis effect	[97]

## Data Availability

Not applicable.

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
