# Peer review of "Mesoporous Nanoparticles for Diagnosis and Treatment of Liver Cancer in the Era of Precise Medicine"

_pharmaceutics, 2022, doi:10.3390/pharmaceutics14091760_

Round 1

Reviewer 1 Report

The manuscript focus on the use of mesoporous nanoparticles for diagnosis and treatment of liver cancer. Overall it is well written, organized and up to date. It would be nice to if the authors could provide some original images rather than be mostly taken from other review papers. Most importantly, a section focusing the toxicity of the suggested systems is of main importance due to their bioaccumulation in the liver, and should be added to the manuscript. Some other minor issues has to be addressed: 

-The text of caption of figure 1 should be moved to the main text.

-The figures 2 and 3 have very poor definitions. Better ones have to be provided.

-The references are placed together with the last word of each sentence. A space between has to be placed.

-The conclusions section should be shortened and without references.

Author Response

We really thank you for your professional and careful comments, which is very helpful to revise our manuscript. We revised the figure 1 according to your suggestions and added table 1 to our manuscript, both of which are the original images produced by our team. The following is the point-to-point answer to your comments. Thank you again for your careful review.

Comment 1: The text of caption of figure 1 should be moved to the main text.

Response:

We have moved the caption of figure 1 to the main text. Please check the highlighted text with yellow color in the revised manuscript.

Comment 2: The figures 2 and 3 have very poor definitions. Better ones have to be provided.

Response:

We tried to modify and add comments to Figures 2 and 3 to make the images easier for readers to understand. Thank you for your advice.

Comment 3: The references are placed together with the last word of each sentence. A space between has to be placed.

Response:

Thank you for your professional comment. We have modified the citation format according to the requirements of the journal contribution.

Comment 4: The conclusions section should be shortened and without references.

Response:

Here is our mistake in drawing up the title of the paragraph. We wanted to discuss the advantages and disadvantages of mesoporous materials here according to the text structure. We thank you for your careful review and have renamed this section as "Discussion and Outlook" and revised the content.

Reviewer 2 Report

1. Please improve abstract and conclusion.

2. References need to be past 5 years unless important.

3. Manuscript Need table on its application in vivo animal model.

4. The clinical applications and studies are missing.

5. The English needs to be checked and corrected by a native English writer.

6. Manuscript Need table on important section (Mesoporous nanocomposites mediated local therapy, and ….)

Author Response

We really thank you for your professional and careful comments, which is very helpful to revise our manuscript. The following is the point-to-point answer to your comments. Thank you again for your careful review.

Comment 1: Please improve abstract and conclusion.

Response:

Thanks for your suggestions. We have carefully modified the abstract and discussion parts. Please find these changes in the yellow section of the revised document.

Comment 2: References need to be past 5 years unless important.

Response:

We updated some of the references and deleted some invalid references from 5 years ago. Thank you for the advice.

Comment 3: Manuscript Need table on its application in vivo animal model.

Response:

Thanks for your professional advice. We have reorganized relevant contents of in vivo experiments and summarized them in Table 1.

Comment 4: The clinical applications and studies are missing.

Response:

Thank you for your forward-looking suggestions. We try to add discussion and reflection on clinical transformation in each section. However, since these studies have not yet been published for translational application, a new component has not been added.

Comment 5: The English needs to be checked and corrected by a native English writer.

Response:

We have invited a native speaker to modify and polish the language of the article. Thank you for your suggestion.

Comment 6: Manuscript Need table on important section (Mesoporous nanocomposites mediated local therapy, and ….)

Response:

Thank you for your advice. We carefully selected high-quality mesoporous studies in the past two years and aggregated these therapeutic studies into Table 1. We hope this conclusion can help readers understand relevant knowledge and arouse thinking.

Reviewer 3 Report

In this review, the authors summarized the advantages and disadvantages of mesoporous nanoparticles for diagnosis and treatment of liver cancer in the era of precise medicine. The article is attention-grabbing for readers but has some points that should be explained.

  1. The author should mention the mesoporous nanoparticles' definition and composition before listing their application. Also, a simple comparison between the mesoporous nanoparticles and the regularly used nanoparticles
  1. In line 87, the author wrote, "Mesoporous silica nanoparticles, represented by mesoporous silica nanoparticles (MSNs), have a simple synthesis method." The mesoporous silica nanoparticles are repeated. It must be written as "Mesoporous silica nanoparticles (MSNs) have a simple synthesis method."

  1. Some abbreviations must be defined as they may confuse the reader.

Author Response

Comments:

In this review, the authors summarized the advantages and disadvantages of mesoporous nanoparticles for diagnosis and treatment of liver cancer in the era of precise medicine. The article is attention-grabbing for readers but has some points that should be explained.

The author should mention the mesoporous nanoparticles' definition and composition before listing their application. Also, a simple comparison between the mesoporous nanoparticles and the regularly used nanoparticles

In line 87, the author wrote, "Mesoporous silica nanoparticles, represented by mesoporous silica nanoparticles (MSNs), have a simple synthesis method." The mesoporous silica nanoparticles are repeated. It must be written as "Mesoporous silica nanoparticles (MSNs) have a simple synthesis method."

Some abbreviations must be defined as they may confuse the reader.

Response:

Thank you for your professional review. We have added tables and pictures to enrich this point of view. The differences between mesoporous nanoparticles and other types of nanoparticles are also discussed.

Thank you for pointing out the error on line 87, which we have corrected following your advice.

Finally we went over the abbreviations in case anything was missing.

Round 2

Reviewer 1 Report

The authors have addressed all the suggestions so teh manuscript is suitable for publication now.

Author Response

Thank you for your approval of our review article.

Reviewer 2 Report

1. Please improve introduction.

2. References need to be past 5 years unless important.

3. Manuscript Need table on its application in vitro. It is very important.

4. Table 1, should be improved.

5. The clinical applications and studies are missing.

6. The English needs to be checked and corrected by a native English writer.

7. Authors should be added future perspective in manuscript.

Author Response

Response to Reviewer 2:

Comment 1: Please improve introduction.

Response:

Thank you for your suggestion. We have further modified our introduction in manuscript R2. (Page 1)

Comment 2: References need to be past 5 years unless important.

Response:

Thanks a lot. We have deleted all the reference before the year 2017. Thank you for your careful review. (Page 17)

Comment 3: Manuscript Need table on its application in vitro. It is very important.

Response:

Thanks for the comment. We have modified the table in our manuscript R2 about the application of mesoporous nanoparticles in vivo. It may be pointless to list in vitro experiments separately, since most studies have been done the relative validation both in vitro and in vivo, and adding a table of in vitro experiments would be overlapped. In addition, compared with the preliminary results obtained from in vitro experiments, the results of in vivo experiments are more meaningful and valuable for transformation. (Page 12)

Comment 4: Table 1, should be improved.

Response:

Done. We checked the table 1 and modified the content where was necessary. (Page 12)

Comment 5: The clinical applications and studies are missing.

Response:

Thanks again. This paper mainly discusses and summarizes the basic related research. In fact, the clinical application and related tests of mesoporous materials are also lacking at present. As we mentioned in the discussion, there have been no trials of this kind in liver cancer patients, and there may be some ethical reasons for this. Thank you for your understanding and support. (Page 3)

Comment 6: The English needs to be checked and corrected by a native English writer.

Response:

Thanks for the comment. The language has been modified by a native speaker. If there were still some problems, please do not hesitate to contact us to further revise.

Comment 7: Authors should be added future perspective in manuscript.

Response:

According to your suggestion, we try to further summarize and outlook this part of the work in the discussion section. Thank you for your careful review. (Page 15)

Round 3

Reviewer 2 Report

All the said comments have been done completely and manuscript accept in present form.

Author Response

Thank you for your careful review and accept of our manuscript.